# Advancing Precision Oncology in Hereditary Paraganglioma-Pheochromocytoma Syndromes: Integrated Interpretation and Data Sharing of the Germline and Tumor Genomes

**DOI:** 10.3390/cancers16050947

**Published:** 2024-02-26

**Authors:** Huma Q. Rana, Diane R. Koeller, McKenzie Walker, Busra Unal, Alison Schwartz Levine, Anu Chittenden, Raymond A. Isidro, Connor P. Hayes, Monica D. Manam, Ryan M. Buehler, Danielle K. Manning, Justine A. Barletta, Jason L. Hornick, Judy E. Garber, Arezou A. Ghazani

**Affiliations:** 1Division of Cancer Genetics and Prevention, Dana-Farber Cancer Institute, Boston, MA 02215, USAanu_chittenden@dfci.harvard.edu (A.C.);; 2Division of Population Sciences, Dana-Farber Cancer Institute, Boston, MA 02215, USA; 3Harvard Medical School, Boston, MA 02115, USA; 4Division of Genetics, Brigham and Women’s Hospital, Boston, MA 02115, USAbusraunalbav@gmail.com (B.U.);; 5Department of Pathology, Brigham and Women’s Hospital, Boston, MA 02115, USA; 6Department of Medicine, Brigham and Women’s Hospital, Boston, MA 02115, USA

**Keywords:** INT^2^GRATE Oncology Consortium, somatic and germline integration, INT^2^GRATE, tumor signature profile, germline VUS, hereditary paraganglioma–pheochromocytoma syndromes (HPPGLs), PGL, PCC

## Abstract

**Simple Summary:**

The standard interpretation methods of germline variants in cancer are limited due to overlapping features in constitutional and sporadically derived forms of cancers and the unavailability of key differentiating details in public settings. To address this challenge, we established INT^2^GRATE (INTegrated INTerpretation of GeRmline And Tumor gEnomes), a multi-institution oncology consortium to advance the integrated application of constitutional and tumor data and share the integrated variant data in a publicly accessible knowledgebase. The aim of our study is to introduce INT^2^GRATE|HPPGL, a platform for the integrated interpretation of hereditary paraganglioma–pheochromocytoma syndromes (HPPGL). We describe the details of the INT^2^GRATE|HPPGL Variant Evidence Framework for succinate dehydrogenase (SDHx) genes using key HPPGL personal and family history, as well as tumor-derived evidence. We applied the INT^2^GRATE|HPPGL Variant Evidence Framework to 8600 variants to programmatically process and share the integrated variant data in ClinVar using a custom-made INT^2^GRATE variant submission pipeline. This novel integrated variant assessment and data sharing in hereditary cancers is essential to help improve the clinical interpretation of genomic variants and advance precision oncology.

**Abstract:**

Standard methods of variant assessment in hereditary cancer susceptibility genes are limited by the lack of availability of key supporting evidence. In cancer, information derived from tumors can serve as a useful source in delineating the tumor behavior and the role of germline variants in tumor progression. We have previously demonstrated the value of integrating tumor and germline findings to comprehensively assess germline variants in hereditary cancer syndromes. Building on this work, herein, we present the development and application of the INT^2^GRATE|HPPGL platform. INT^2^GRATE (INTegrated INTerpretation of GeRmline And Tumor gEnomes) is a multi-institution oncology consortium that aims to advance the integrated application of constitutional and tumor data and share the integrated variant information in publicly accessible repositories. The INT^2^GRATE|HPPGL platform enables automated parsing and integrated assessment of germline, tumor, and genetic findings in hereditary paraganglioma–pheochromocytoma syndromes (HPPGLs). Using INT^2^GRATE|HPPGL, we analyzed 8600 variants in succinate dehydrogenase (SDHx) genes and their associated clinical evidence. The integrated evidence includes germline variants in SDHx genes; clinical genetics evidence: personal and family history of HPPGL-related tumors; tumor-derived evidence: somatic inactivation of SDHx alleles, *KIT* and *PDGFRA* status in gastrointestinal stromal tumors (GISTs), multifocal or extra-adrenal tumors, and metastasis status; and immunohistochemistry staining status for *SDHA* and *SDHB* genes. After processing, 8600 variants were submitted programmatically from the INT^2^GRATE|HPPGL platform to ClinVar via a custom-made INT^2^GRATE|HPPGL variant submission schema and an application programming interface (API). This novel integrated variant assessment and data sharing in hereditary cancers aims to improve the clinical assessment of genomic variants and advance precision oncology.

## 1. Introduction 

Hereditary paraganglioma–pheochromocytoma syndromes (HPPGLs) refer to a group of conditions characterized by the presence of sympathetic or parasympathetic paragangliomas (PGLs). PGLs are tumors that arise from neural crest-derived progenitors, and pheochromocytomas (PCCs) are PGLs confined to the adrenal medulla. The clinical diagnosis of HPPGL is established by the presence of a personal and family history of PGL or PCC and a heterozygous germline pathogenic variant in HPPGL-related genes identified by laboratory genetic testing [1]. 

Understanding the complexities of the genetic background in HPPGL is important. Affecting about 0.6/100,000 individuals/year [2], PGLs and PCCs are rare and highly heritable tumors. An estimated 35–40% of cases are associated with a hereditary predisposition [2]. Germline variants in HPPGL-related genes have been reported in up to 79% of PGL/PCC patients with a positive family history of non-syndromic PCC/PGL and 54% in patients with head and neck PGL [3], suggesting possible confounding genetic modifiers and incomplete penetrance. Most PGL/PCC tumors are reportedly sporadic (i.e., not inherited) but present characteristic features similar to inherited tumors [1].

Heterozygous germline pathogenic variants in the tumor suppressor genes *SDHA*, *SDHB*, *SDHC*, and *SDHD* cause HPPGL syndrome. These genes, collectively known as the SDHx genes, encode the four subunits of the mitochondrial enzyme succinate dehydrogenase, which is involved in the citric acid cycle and the electron transport chain [4]. In addition to SDHx genes, *VHL*, *NF1*, *RET*, *EGLN1*, *FH*, *KIF1B*, *MAX*, *MEN1*, *SDHAF2*, and *TMEM127* genes can result in overlapping PGL/PCC phenotypes but are related to other well-known cancer susceptibility syndromes. These observations highlight the need for a comprehensive assessment of focused and clinically related evidence associated with germline variants in HPPGL without confounding factors. This need is further amplified as the size of gene panels in genetic testing is expanding and the size of incidental findings with uncertain significance is increasing. 

We have previously demonstrated the utility of tumor-derived and clinical genetic information in elucidating the clinical relevance of germline variants in selected cancers [5,6,7,8]. INT^2^GRATE (INTegrated INTerpretation of GeRmline And Tumor gEnomes) is a multi-institution oncology consortium. The aim of INT^2^GRATE is twofold: bringing together constitutional and relevant tumor-derived information in cancer syndromes and sharing the integrated variant data publicly. In this paper, we describe the development of the INT^2^GRATE|HPPGL platform, the genetic parameters, and their utility in the integrated assessment of germline and tumor-derived evidence in HPPGL. We also describe the systematic collation and processing of 8600 SDHx variants using INT^2^GRATE|HPPGL and the subsequent data sharing by submitting these variants directly from INT^2^GRATE|HPPGL to ClinVar. This novel method of processing and sharing comprehensive integrated germline and somatic evidence is a crucial step in advancing precision oncology in genomics. 

## 2. Methods

### 2.1. Development of INT^2^GRATE|HPPGL Platform

The INT^2^GRATE|HPPGL platform has four components: a variant evidence framework, a Web-based application for user-friendly single-variant processing, an automated system for parsing batched variant data, and an application programming interface (API) for programmatic data sharing. Each component is described separately below. 

#### 2.1.1. INT^2^GRATE|HPPGL Variant Evidence Framework

The INT^2^GRATE|HPPGL Variant Evidence Framework was developed using four types of well-established clinical evidence routinely used in the assessment and management of patients with HPPGL: (1) germline variants in SDHx genes; (2) patient-derived, clinical genetics data relevant to HPPGL; (3) tumor-derived genetic data; (4) and protein expression of SDHA and SDHB as assessed by immunohistochemistry (IHC). To assess the utility of the Variant Evidence Framework, an expert group was formed comprising board-certified medical geneticists with experience in cancer genetics and tumor profiling, board-certified clinicians and genetic counselors with expertise in hereditary cancer diagnostics including HPPGL, and board-certified pathologists with expertise in endocrine pathology and surgical pathology including IHC. The description and rationale for each parameter are described in the Results section. The Variant Evidence Framework marks the presence or absence of each variant parameter, assesses the combination of this evidence, and catalogs different scenarios separately for each gene (Table 1).

#### 2.1.2. INT^2^GRATE Digital Web Portal for Single Variants

The Web portal was developed and made publicly available for processing of single variants according to the INT^2^GRATE|HPPGL Variant Evidence Framework. The INT^2^GRATE Web-based portal has a user interface and back-end programming. Each component is described below. The INT^2^GRATE website is available at https://int2grate.bwh.harvard.edu (accessed on 17 July 2023).

##### User Interface (UI)

A Web application user interface (UI) was developed for INT^2^GRATE|HPPGL to facilitate the ease of use of the Variant Evidence Framework (Figure 1). The UI shows the INT^2^GRATE|HPPGL parameters as a one-page questionnaire with user single-response radio button inputs. Text fields were created for variant information input (i.e., HGVS, cDNA, and protein). User responses are programmatically assessed using a Javascript (Oracle Corporation, Austin, TX, USA) algorithm that functions in the user’s browser to align the user’s responses with the corresponding Variant Evidence Framework categories and responses in the backend of the website. User data are processed on the client side in the user’s browser and are not stored on the host server. Upon completion of the answers (i.e., the input of the HPPGL’s related parameters) in the questionnaire and after submitting the form, the user can see the corresponding INT^2^GRATE|HPPGL comment. The UI provides an option for the user to download and store a spreadsheet documenting their responses and associated INT^2^GRATE|HPPGL comments. Additionally, the UI features an API for data sharing and automatic submission of variants and associated INT^2^GRATE|HPPGL parameters to ClinVar (as discussed in Section 2.1.4. Variant Data Sharing via API below). 

##### INT^2^GRATE Backend

Javascript algorithms operate in the backend of the INT^2^GRATE|HPPGL site to receive the responses from the UI questionnaire form. User input is passed into a series of conditional logic statements to determine if the combination of selections conforms to a scenario delineated by the INT^2^GRATE|HPPGL Variant Evidence Framework. If the combination of responses matches those within a unique INT^2^GRATE|HPPGL Variant Evidence Framework option, the corresponding comment and explanation are returned to the user in the UI. If the user’s combination of responses is outside scenarios accounted for in the INT^2^GRATE|HPPGL Variant Evidence Framework, a message notifies the user that their combination of responses is not currently accounted for within the Variant Evidence Framework. 

#### 2.1.3. INT^2^GRATE|HPPGL Automated Batch Variant Processing

This feature of the INT^2^GRATE|HPPGL platform allows for automated parsing and analysis of variants on a large scale. Relevant clinical genetics, germline variants, and tumor-derived genetic and IHC data files for each patient are mapped to create a comprehensive Integrated Somatic and Germline variant Database (ISGD) for HPPGL data. The Python (Python Software Foundation, Wilmington, DE, USA) program reads the data from ISGD and programmatically conforms them to the parameters defined in the INT^2^GRATE|HPPGL Variant Evidence Framework. A built-in process performs the quality assessment. The program then automatically processes the variant data according to the INT^2^GRATE|HPPGL Variant Evidence Framework and produces the INT^2^GRATE|HPPGL codes before creating a knowledgebase of INT^2^GRATE|HPPGL variants. Processing a new batch of variants automatically adds them to the INT^2^GRATE|HPPGL knowledgebase for data sharing (e.g., ClinVar API batch submission as discussed in Large-Scale INT^2^GRATE Variant Submission below) or research.

#### 2.1.4. INT^2^GRATE|HPPGL Variant Data Sharing via API

To facilitate data sharing and variant submission to ClinVar, INT^2^GRATE offers two routes to interface with the ClinVar API for programmatic variant submission. The INT^2^GRATE Oncology Consortium is a registered submitter organization in ClinVar. 

##### INT^2^GRATE UI Variant Submission 

This option is for single-variant submissions by registered users of the INT^2^GRATE web portal. User responses from the INT^2^GRATE|HPPGL Web-based questionnaire are conformed to an API submission schema. A confirmation popup allows the user to review the content before submission. User data are then automatically formatted in a standardized format and sent to ClinVar. The user submits the variant to ClinVar using a password-protected submission.

##### Large-Scale INT^2^GRATE Variant Submission 

For batch submission of variant data, an automatic pipeline was developed in Python for the automatic and simultaneous processing of thousands of variants from the INT^2^GRATE|HPPGL knowledgebase (Section 2.1.3 in the Methods). Variants and associated evidence are automatically structured to conform to the custom-made INT^2^GRATE|HPPGL variant submission schema. The program reads each line of the submission database and loops through all the variant data to send the information to ClinVar. This process includes performing a QC step and creating a submission log for each variant as it iterates over each line. To ensure patient privacy, the ISGD, as well as variant submission preparation and submission processes, includes no protected health information (PHI) at any step. 

### 2.2. Germline Genetic Laboratory Testing 

Germline genetic testing was completed in commercial laboratories (Invitae, San Francisco, CA, USA; Ambry Genetics, Aliso Viejo, CA, USA) as described previously. Briefly, DNA samples from peripheral blood or saliva were enriched for targeted regions using a hybridization-based protocol. Sequencing was performed using a next-generation sequencing (NGS) platform. Copy number alterations in exonic regions were processed by analysis of the read depth for each target sequence, mean read depth, and read-depth distribution obtained from parameters in validation experiments.

### 2.3. Tumor Genetic Laboratory Testing

Tumor testing was performed by the OncoPanel test at the Center for Advanced Molecular Diagnostics (CAMD) laboratory in the Pathology Department at Brigham and Women’s Hospital (BWH, Boston, MA, USA) as part of a routine clinical test and as previously described [8].

#### 2.3.1. Single-Nucleotide Variant (SNV)/Indel Analysis 

SNV calling was performed with MuTect (Broad Institute, Cambridge, MA, USA) and GATK Indelocator (Broad Institute, Cambridge, MA, USA) to identify somatic SNVs and indels, respectively, as previously described [5,7]. Variants were filtered based on presence in the Exome Sequencing Project (ESP) and/or gnomAD databases with an allele frequency > 0.1% in any sub-population. Variants were also filtered if detected in the plate normal control run with each assay. Any variant filtered by those criteria but present in the COSMIC database (COSMIC, Wellcome Sanger, London, UK) at least twice was subsequently rescued. Each variant was annotated with the gene, genome coordinates, reference and alternate alleles, coverage, allele fraction, cDNA, and protein change. Based on the somatic OncoPanel validation data, the limit of detection was determined to be 50× coverage and 10% variant allele fraction. Variants with lower coverage and/or allele fraction or with fewer than five unique reads of support were excluded from the analysis. 

#### 2.3.2. OncoPanel Copy Number Analysis

RobustCNV, developed at the Dana-Farber Cancer Institute (DFCI, Boston, MA, USA), was used to detect somatic copy number variants (CNVs), as described previously [5,7]. Each baited genomic segment was normalized against the panel of normal samples. Segments with a Log2 ratio on the zero line were considered neutral copy numbers. Copy numbers with a designation of low amplification, high amplification, one copy deletion, or two copy deletions were recorded before technical review. Low-level amplifications were called at a Log2 ratio ≥ 0.43 and losses at a Log2 ratio ≤ −0.32. 

#### 2.3.3. OncoPanel Structural Analysis

BreaKmer, developed at DFCI (DFCI, Boston, MA, USA), was used to detect somatic chromosomal rearrangements, large indels, and inversions, as described previously [5,7,9]. The tool identifies sequence fragments that map to non-contiguous regions of the reference sequence. Structural variant (SV) fragments were presented with the gene(s) involved, genome coordinates, and an IGV snapshot for visual confirmation. Split reads (a single read that maps to two regions of the genome) and discordant read pairs (the paired ends of a sequenced read map to different genomic locations) were called by the pipeline. Variants with ≤2% total split and discordant reads/total coverage across the detected breakpoints were closely reviewed. SV calls in repetitive genomic regions were excluded from the analysis.

### 2.4. Laboratory Testing—Immunohistochemistry (IHC) 

For cases stained at BWH, IHC for SDHA and/or SDHB proteins was performed as previously described [8]. Briefly, slides with formalin-fixed, paraffin-embedded tissue sections were deparaffinized, rehydrated, subjected to antigen retrieval (pressure cooker; pH, 6.1; citrate buffer; Target Retrieval Solution, DAKO, Santa Clara, CA, USA), and incubated with the primary antibodies targeting SDHA (1:800, clone 2E3; ab14715, Abcam, Chuo City, Tokyo) or SDHB (1:300, clone 21A11; ab14714, Abcam, Chuo City, Tokyo). This was followed by secondary detection with the DAKO Envision plus system (DAKO, Santa Clara, CA, USA). 

### 2.5. Patient Cohorts and Data Query

A total of 8600 patient data were retrospectively collected for this study from 2018–2022. An agnostic approach was used to ensure the collection of SDHx-positive cases without prior knowledge of HPPGL evaluation to allow for the processing of all variants indiscriminately through the INT^2^GRATE|HPPGL process. The patient inclusion criteria were the presence of at least one germline or somatic variant in the *SDHA, SDHB, SDHC*, or *SDHD* genes. The query pulled data from 8106 patients with a reported SDHx alteration identified by the OncoPanel test at BWH. Separately, a total of 637 patients were collected from DFCI’s Center for Cancer Genetics and Prevention whose germline testing included at least one variant in any SDHx gene. The merging of these two databases identified 109 patients who had both germline testing and OncoPanel tumor testing (Cohort 1), 494 patients who had only germline testing (Cohort 2), and 7997 who had only OncoPanel tumor testing (Cohort 3). All data parameters for the INT^2^GRATE|HPPGL Variant Evidence Framework were collected for patients in the three patient cohorts when available. 

Germline variants were collected using a custom query for all SNVs, indels, or copy numbers (CNs) that were reported as pathogenic, likely pathogenic, VUS, or common polymorphism in SDHx genes as part of routine patient care. For patients with any reported variant in SDHx genes, all other germline findings were queried to assess the presence of other HPPGL-related genes. A comprehensive clinical history was collected to include personal and family history of cancer; parent of origin for *SDHD* variants; the presence of multiple, multifocal, or extra-adrenal tumors; and the presence of PGL/PCC metastasis. 

Tumor OncoPanel data were collected using a custom query for all cases that have reported variants in any SDHx genes. Somatic variants included SNVs, indels, CNVs, or SVs that were reported as part of routine tumor profiling. For SDHx-positive cases identified by OncoPanel, *KIT* and *PDGFRA* variant data, as well as IHC findings, were collected. 

## 3. Results

### 3.1. Development of INT^2^GRATE|HPPGL Variant Evidence Framework and Rationale 

The INT^2^GRATE|HPPGL Variant Evidence Framework utilizes four types of evidence routinely used to assess HPPGL patients (Table 1). 

#### 3.1.1. Germline Variant in SDHx Genes and Rationale 

Alterations in the SDHx complex produce well-known IHC staining patterns for SDHA and SDHB (described in the IHC section below). Also, SDHx genes are tumor suppressors associated with loss of function and loss of heterozygosity (LOH). Therefore, these genes were included in the INT^2^GRATE|HPPGL Variant Evidence Framework. Other genes associated with PGL risk include *SDHAF2*, *MAX*, *TMEM127*, *EPAS1*, *VHL*, *RET*, and *NF1*. These genes were not included in the INT^2^GRATE Variant Evidence Framework, as they are not associated with clinically informative tumor-derived IHC findings and/or tumor suppressor tumor signatures. The assessment of these genes and overlapping clinical features was outside the scope of the present work. The INT^2^GRATE|HPPGL Variant Evidence Framework requires the presence of only one germline variant in one SDHx gene, as multiple germline variants in SDHx genes or other genes associated with HPPGL syndrome would call into question the significance of the germline variant that is being assessed. These conditions were selected to ensure a conservative approach in assessing one informative germline variant at a time (Table 1). 

#### 3.1.2. Clinical Genetics Criteria and Rationale

The evaluation and inclusion of clinical genetics criteria are requirements in the design of INT^2^GRATE|HPPGL. The presence of a strong personal and/or family history with concordant tumor data is highly indicative of HPPGL syndrome. This requirement for a strong personal and/or family history was included to flag germline alterations with potential incomplete penetrance or variants in sporadic SDHx-deficient tumors. In the clinical criteria category, three separate conditions were used in the INT^2^GRATE|HPPGL Variant Evidence Framework (Table 1): 

##### Personal History

HPPGL syndrome is primarily associated with an increased risk of PGL and PCC with/without associated gastrointestinal stromal tumor (GIST), renal cell carcinoma (RCC), and pituitary adenomas. Since PGL/PCC ± associated GIST is rare and more strongly associated with HPPGL, it was included as a personal history criterion for evaluation. GIST is more common in individuals with pathogenic variants (PVs) in *SDHC* and *SDHA* compared to *SDHB* and *SDHD*. RCCs and pituitary adenomas are common and not specific to HPPGL; therefore, they were not included criteria in the conservative INT^2^GRATE|HPPGL Variant Evidence Framework. 

##### Family History

HPPGL is highly indicated in patients with a PGL or PCC with relatives with a PGL, PCC or GIST, multiple PGLs (including bilateral PCCs), multifocal tumors with synchronous or metachronous tumors, or metastatic PGL/PCC [10]. Therefore, these family history factors were included in the INT^2^GRATE Variant Evidence Framework. Multiple or metastatic PGL/PCC is more common in individuals with a PV in SDHB or a paternally inherited PV in SDHD compared the those with a PV in SDHA or SDHC.

##### Parent of Origin

Paternal inheritance was included in the evaluation of variants in the SDHD gene, as only paternally inherited PVs in SDHD are known to be associated with a high risk of developing a component tumor. Therefore, paternal inheritance was used as a required factor in the INT^2^GRATE|HPPGL Variant Evidence Framework for germline variants in the SDHD gene.

#### 3.1.3. Tumor-Derived Genetic Information and Rationale 

The somatic SDHx allele, as well as *KIT* and *PDGFRA* mutational status, was used as a genomic parameter in the INT^2^GRATE|HPPGL Variant Evidence Framework. Given the rarity of multiple, multifocal, extra-adrenal, or metastasis PGL/PCC, these parameters were only used for evidence gathering (Table 1):

##### Somatic Inactivating Allele

SDHx genes are tumor suppressors and therefore are inactivated through a two-hit model. INT^2^GRATE Variant Evidence Framework requires the presence of a second hit or a somatic inactivating allele in the particular SDHx gene, which is detected on tumor sequencing and is absent on germline sequencing. Biallelic somatic inactivating alleles in the SDHx genes may provide a non-germline (i.e., sporadic) explanation for tumor development. 

##### KIT or PDGFRA Mutational Status

Somatic activating mutations in *KIT* and *PDGFRA* are the most common causes of sporadic GIST [11]. *KIT* or *PDGFRA* mutations should be absent from SDH-deficient GIST [12]. SDH-deficient GISTs arise in the stomach, account for 5–7.5% of GISTs, and have distinct clinical and morphological features [13]. Therefore, the wild-type genetic status for both *KIT* and *PDGFRA* was included as a required tumor parameter in the INT^2^GRATE|HPPGL Variant Evidence Framework.

##### Multiple, Multifocal, or Extra-Adrenal Tumors and PGL/PCC Metastasis

Individuals with HPPGL are more likely to have multiple, multifocal, and/or metastatic tumors compared to patients with sporadic PGL/PCC [14,15]. Extra-adrenal paragangliomas are associated with a higher incidence of cancer metastasis [16] more commonly associated with *SDHB* mutation. Currently, the exact association of these categories of tumors with germline status is not know; therefore, they were not used as main parameters in the design of the Variant Evidence Framework. They were included as ancillary evidence to help collect associated evidence. 

#### 3.1.4. Tumor Immunohistochemistry Pattern and Rationale 

Immunohistochemistry for the SDHB protein is routinely used to screen cases of PGL/PCC and triage for genetic testing. It is also used to further evaluate *KIT* and *PDGFRA*-wild-type gastric GIST. Loss of any of the components of the SDHx complex results in loss of SDHB staining [17,18,19]; therefore, loss of SDHB expression indicates biallelic inactivation of one of the SDHx component proteins. Often, if SDHB is lost, SDHA staining is performed sequentially. Loss of SDHA has high specificity for an underlying alteration in *SDHA*. Therefore, the patterns of SDHA and SDHB IHC were included in the INT^2^GRATE|HPPGL Variant Evidence Framework (Table 1). 

### 3.2. Assignment of INT^2^GRATE|HPPGL Categories 

The cumulation of this variant evidence presents different but common or relatively common scenarios observed in the clinical assessment of HPPGL patients. Each scenario is tagged with an INT^2^GRATE|HPPGL internal code, a category, and a comment describing the findings associated with that scenario (Table 1). The INT^2^GRATE POSITIVE category is marked for scenarios where all evidence parameters are conservatively expected in HPPGL cases. The INT^2^GRATE NEGATIVE category denotes cases where all evidence congruently shows negative findings (i.e., pattern not consistent with HPPGL). All other scenarios are denoted as INT^2^GRATE NEUTRAL (Table 1). 

### 3.3. Application of INT^2^GRATE|HPPGL 

After developing the INT^2^GRATE|HPPGL platform, we ran the SDHx cases to systematically collate the variant data and associated evidence according to the Variant Evidence Framework and programmatically share the data in ClinVar. A total of 8049 somatic and 603 germline SDHx variants were collected and analyzed according to the INT^2^GRATE|HPPGL Variant Evidence Framework. For germline variants, the assessment included the presence of an additional SDHx variant in the same patient; the presence of any additional variants in HPPGL-associated genes; personal history of PGL/PCC-associated tumors; family history and relatives with PGL/PCC/GIST; reports of multiple, multifocal, or extra-adrenal tumors; and the presence of PGL/PCC metastasis. For tumor-derived data, somatic variants in SDHx genes were evaluated to determine the inactivating allele status (i.e., inactivating SNV, CNV, and/or SV). *KIT* and *PDGFRA* variants were evaluated to indicate wild-type status. For cases where IHC was indicated, data for SDHB and SDHA were evaluated. 

### 3.4. Patient Clinical Presentations 

Patients in Cohort 1 (*n* = 109) were 33% male and 72% female, with a median age at cancer diagnosis of 47 (2–84 years old). In this Cohort, 27% (*n* = 30) had a personal history of HPPGL-related cancers, and 0.03% had a family history of HPPGL-related cancers (Table 2, Appendix A). Cohort 2 (*n* = 494) was 29% male and 71% female; 7% and 2% presented with a personal and family history of HPPGL-related cancers, respectively (Table 3). Patients in Cohort 3 (*n* = 7997) had only OncoPanel and tumor data; they were not evaluated for any familial cancer and did not have germline test results. 

### 3.5. INT^2^GRATE|HPPGL Variant Analysis 

Cohort 1

Cohort 1 has the most comprehensive collection of both germline and somatic data. In Cohort 1 (*n* = 109), 37 patients were positive for a pathogenic or likely pathogenic germline variant in SDHx genes based on the standard ACMG classification. INT^2^GRATE|HPPGL analysis showed that about 46% (16/37) of patients had a personal history of PGL/PCC (with or without associated GIST), and only 14% (5/37) had a family history of PGL/PCC (Table 4). About 35% (13/37) of cases had a somatic inactivating allele consistent with loss of heterozygosity of the SDHx gene being assessed (the somatic allele harbored one copy loss in 11/37 cases and an inactivating SNV in 2/37 of cases). Of these cases with an apparent LOH, only eight showed an IHC pattern consistent with biallelic inactivation of the relevant SDH subunit gene. Overall, 23/37 were negative for a somatic inactivating SNV, copy number, or structural alteration. 

Of all cases with a pathogenic or likely pathogenic germline SDHx variant, only three cases (subjects 33, 61, and 82) had all parameters of the INT^2^GRATE|HPPGL Variant Evidence Framework. This observation not only highlights the conservative approach in the design of the INT^2^GRATE|HPPGL Variant Evidence Framework but also underscores the degree of variable expressivity in the HPPGL-related phenotypes. 

The remaining cases in Cohort 1 had SDHx variants classified as VUS by commercial laboratories. INT^2^GRATE|HPPGL analysis of cases with *SDHA* VUS (*n* = 40) showed that four patients had a personal history of PGL/PCC ± associated GIST, none had a family history of PGL/PCC, none carried a somatic inactivating allele, none had any IHC findings suggestive of SDHA/SDHB loss of expression, and all had a wild-type *KIT* or *PDGFRA* status (Table 5). Similarly, INT^2^GRATE|HPPGL analysis of cases with *SDHB* variants (*n* = 15) were negative for personal (except for one) and family history of PGL/PCC ± associated GIST; somatic inactivating alterations in *SDHB* (except for one); multiple, multifocal, or extra-adrenal tumors; and PGL/PCC metastasis (Table 6). With the wild-type status of *KIT* or *PDGFRA* and the absence of abnormal IHC patterns, these variants are unlikely to be involved in HPPGL. Except for patient #3, with a diagnosis of PGL and GIST, all patients in this group were diagnosed with cancers other than neuroendocrine or any PGL/PCC component tumors (Table 6). This combination of evidence does not provide strong support in favor of HPPGL, particularly as pathogenic germline *SDHB* variants are known to be highly penetrant. INT^2^GRATE|HPPGL analysis revealed a similar pattern for cases with variants in *SDHC* (Table 7) and *SDHD* (Table 8), suggesting the VUS in these cases might be an incidental finding. Reporting and analyzing a large INT^2^GRATE|HPPGL variant dataset over time may help with the reclassification of many VUS to likely benign (LB).

Cohort 2

Patients in this group (*n* = 494) were evaluated in genetics but did not have a tumor OncoPanel performed. Despite the lack of tumor genetics data, INT^2^GRATE|HPPGL analysis was informative in collecting evidence related to HPPGL. About 7% (36/494) of patients had a personal history of PGL/PCC ± associated GIST or no multiple, multifocal, or extra-adrenal tumors and no family history of PGL/PCC (Appendix A). Germline variants for all except three patients in this subset were classified as pathogenic or likely pathogenic (P/LP) by the commercial laboratory. The benign variants were *SDHB*:c.487T>C (p.Ser163Pro) in two unrelated individuals with PCC and *SDHB*:c.8C>G (p.Ala3Gly) in one patient with PGL. Of 494 individuals in this group, 20% (108/494) had family members who were also carriers of pathogenic (80%), VUS (14%), and suspected polymorphic (4%) SDHx variants. 

Interestingly, 201 individuals in this Cohort had a reported P/LP germline variant in one SDHx gene, 169 of whom did not have a personal history of PGL/PCC ± associated GIST and multiple, multifocal, or extra-adrenal tumors. Only 7% (11/169) had a family history of PGL, with the remaining 93% having no personal or family history of PGL/PCC tumors. Most of the reported P/LP variants were in *SDHB* (*n* = 84). Despite the expected high penetrance of *SDHB* germline pathogenic variants and a strong association with PGL [16], the personal and family histories of these patients were unremarkable for HPPGL. The remaining reported P/LP variants were in *SDHA* (*n* = 37), *SDHC* (*n* = 35), and *SDHD* (*n* = 11). The parent of origin was not tested for these individuals, except for one individual whose *SDHD*:c.242C>T (p.Pro81Leu) variant showed a maternal mode of inheritance. *SDHD* variants are known to follow a parent-of-origin effect in HPPGL almost exclusively when they are paternally inherited. 

Cohort 3

Patients in Cohort 3 (*n* = 7997) were not evaluated for inherited cancers, but their tumors were tested by OncoPanel. Overall, these cases were positive for at least one genetic alteration (SNV, CN, or SV) in an SDHx gene. The cancer types were diverse. Only a small fraction of cases showed biallelic somatic alteration in *SDHA* (*n* = 11), *SDHB* (*n* = 18), *SDHC* (*n* = 1), and *SDHD* (*n* = 9) (Appendix A). PGL/PCC-related cancers were present only in three cases in *SDHB* showing gastrointestinal stromal tumor, SDH-deficient renal carcinoma, and paraganglioma. Extra-adrenal paraganglioma was present in one case with *SDHC* biallelic variants. The remaining cases with biallelic somatic alteration showed tumor types unrelated to PGL/PCC. 

### 3.6. INT^2^GRATE|HPPGL Variant Submission to ClinVar 

All 8600 cases analyzed by INT^2^GRATE|HPPGL were processed for submission to ClinVar from Cohort 1 (*n* = 109), Cohort 2 (*n* = 494), and Cohort 3 (*n* = 7997). INT^2^GRATE|HPPGL programmatically assessed each category of variants using its automated pipeline, mapped the information contained in the database to conform to the Variant Evidence Framework responses, and generated the associated INT^2^GRATE code. The INT^2^GRATE|HPPGL pipeline then automatically conformed these variants and associated data to our internal custom schema for ClinVar’s API submission designed specifically for the submission by the INT^2^GRATE Oncology Consortium. This information was read into a Python program to mass submit all variants from 8600 cases to ClinVar using the API across the three sample cohorts. 

## 4. Discussion

The interpretation of germline variants in cancer susceptibility genes using standard methods is often limited due to the pleiotropic effect of many cancer genes. ACMG guidelines have been developed for monogenic Mendelian disease, whereas cancers often exhibit more complex genetic disorders. Understanding the role of germline variants in cancer is further confounded by the possible presence of somatically driven tumors and the need to separate constitutional and sporadic genomic findings. Inherited cancers are typically driven using two sets of genomes: the constitutional and somatic genomes. While the constitutional genome is routinely assessed using ACMG criteria, the information from the somatic genome is largely underutilized in this process. Tumors have a wealth of information that may be predictive of cancer behavior and tumor type. Harnessing relevant tumor information can serve as strong evidence to help delineate the function of germline variants in cancer susceptibility genes. We have previously demonstrated the utility of tumor-derived data in the reassessment of the pathogenicity of germline variants in patients with non-syndromic phenotypes [5,6,7,8].

The current limitations in using tumor-derived information in the interpretation of constitutional genetics are threefold. First, detailed tumor-derived information is generally not available in settings in which germline sequence variants are interpreted. Many laboratories or academic institutions do not perform tumor sequencing alongside germline evaluation for each cancer patient. Secondly, details of patient clinical genetics information are not often available to laboratories that run germline sequencing, as only limited information is included on test requisition forms. Lastly, mainly due to the first two limitations, the information about germline variants in public databases does not include informative tumor or clinical genetics findings. The segregation of these two interdependent types of information is a bottleneck in precision oncology and genomics. 

We have addressed this limitation by establishing the multi-institution INT^2^GRATE Oncology Consortium. To that end, we built an infrastructure for the programmatic processing of large-scale variant data for different cancer syndromes, including INT^2^GRATE|HPPGL described here. We also built a companion INT^2^GRATE Web portal for a user-friendly public access application for single-variant use. INT^2^GRATE|HPPGL aims to systematically collect and integrate tumor-derived and clinical genetics information for the assessment of the role of germline SDHx variants and share the integrated germline and tumor information in publicly accessible databases. INT^2^GRATE|HPPGL is not intended to reclassify variants or replace the ACMG assessment criteria. Instead, it is developed to serve as a companion tool to help genetic professionals collect and assess a comprehensive germline and associated tumor set of evidence. Through large-scale variant analysis and data sharing, the true clinical significance of complex variants can be elucidated. 

Here, we performed INT^2^GRATE|HPPGL variant analysis for 8049 somatic and 603 germline SDHx variants and associated clinical evidence in 8600 patients. A larger proportion of our cases did not have germline genetic testing. This observation is consistent with a reported higher proportion of sporadic PGL/PCC cancers compared to the hereditary counterpart [1]. Our study also shows that a small proportion of patients fulfilled clinical suspicion of HPPGL. Of patients in Cohorts 1 and 2 who had detailed germline and clinical genetics findings, 238 individuals had reported pathogenic variants in one SDHx gene, of whom about 80% (190/238) did not have a personal history of PGL/PCC ± associated GIST. This observation can be partly attributed to variable penetrance of SDHx genes, as only *SDHB* and *SDHD* have a high reported penetrance. In our analysis, *SDHB* did not seem to exhibit the expected high penetrance. INT^2^GRATE|HPPGL analysis showed that 47% (89/190) of patients in Cohorts 1 and 2 had a pathogenic *SDHB* variant but no personal history of PGL/PCC ± associated GIST. None of these patients exhibited extra-adrenal sympathetic paragangliomas or metastatic disease, as generally expected of the *SDHB* phenotype [16]. The occurrence of pathogenic variants in *SDHD* with no personal history of PGL/PCC was about 6% (11/190), which is significantly lower than in *SDHB*. Larger INT^2^GRATE|HPPGL datasets can accurately assess the penetrance in SDHx genes. 

INT^2^GRATE|HPPGL variant analysis was also informative for variants that were reported as VUS by commercial laboratories. As shown in Table 5, Table 6, Table 7 and Table 8, in Cohort 1, the cumulation of germline, clinical genetics, tumor genetics, and IHC evidence does not support strong evidence for these variants in HPPGL. Variants with a positive familial carrier status that were not associated with a personal and family history of PGL/PCC may be due to incomplete penetrance or a lack of co-segregation with the disease. Given the absence of consensus for the evaluation of asymptomatic carriers of SDHx variants [20], sharing these variants and their associated INT^2^GRATE evidence in a systematic way in ClinVar can help laboratories and clinical centers with the reclassification of the VUS over time. 

There are limitations in the collection of informative evidence by INT^2^GRATE|HPPGL. The detection of cryptic rearrangements and large structural alterations is not feasible using current sequencing methods. Moreover, epigenetic alterations such as SDHC promoter methylation testing [21] can lead to silencing of the SDHC gene. Because these tests are not routinely evaluated in patients with suspected HPPGL, they were not included as part of the INT^2^GRATE Variant Evidence Framework for HPPGL but may contribute to the loss of function of alleles or loss of heterozygosity in PGL/PCC. Indeed, hypermethylation has been reported in *SDHx*-related tumors as an oncogenic mechanism of SDH inactivation [22,23], and SDHC promoter methylation has been reported in SDHx wild-type PGLs with loss of SDHB by IHC [24]. Furthermore, IHC has some inherent limitations and may not fully capture the molecular complexities of HPPGL. The interpretation of IHC results can be influenced by several factors, including protein lifespan, variant effect on protein structure, weak or diffuse SDHB staining, methodology, and normalization of IHC scores [25]. The intracellular distribution of SDHB IHC staining may vary in tumor cells, leading to complexities or variations in the interpretation of IHC results [26]. IHC at large is an effective tool; however, a lack of concordance between IHC and *SDHD* germline alterations has been reported [27]. Lastly, the different penetrance of SDHx genes and variable expressivity of HPPGL phenotypes can be a confounding factor in variant assessment. The utility of INT^2^GRATE|HPPGL is limited to the current features described here. Any new or evolving genetic landscape of HPPGL or diagnostic and therapeutic strategies may be included in future versions. 

## 5. Conclusions

In conclusion, INT^2^GRATE|HPPGL is a novel platform that systematically collects and integrates germline and somatic variants and associated clinical evidence, conforms data to the Variant Evidence Framework, processes data, and programmatically submits HPPGL variants and associated evidence to ClinVar. INT^2^GRATE variant data help to refine the clinical information that accompanies variant annotation. This clinical context, particularly the information on cancer phenotype and accompanying tumor IHC, provides clinicians who encounter these variants with a level of data not available otherwise. These INT^2^GRATE variant data enable clinicians and laboratories to glean insights and inform decision making around appropriate HPPGL management and international consensus guidelines. The INT^2^GRATE Oncology Consortium is committed to this large-scale, novel, and integrated genomic variation data sharing in a publicly accessible repository to help improve the clinical interpretation of genomic variants and advance precision oncology and research. 

## Figures and Tables

**Figure 1 cancers-16-00947-f001:**
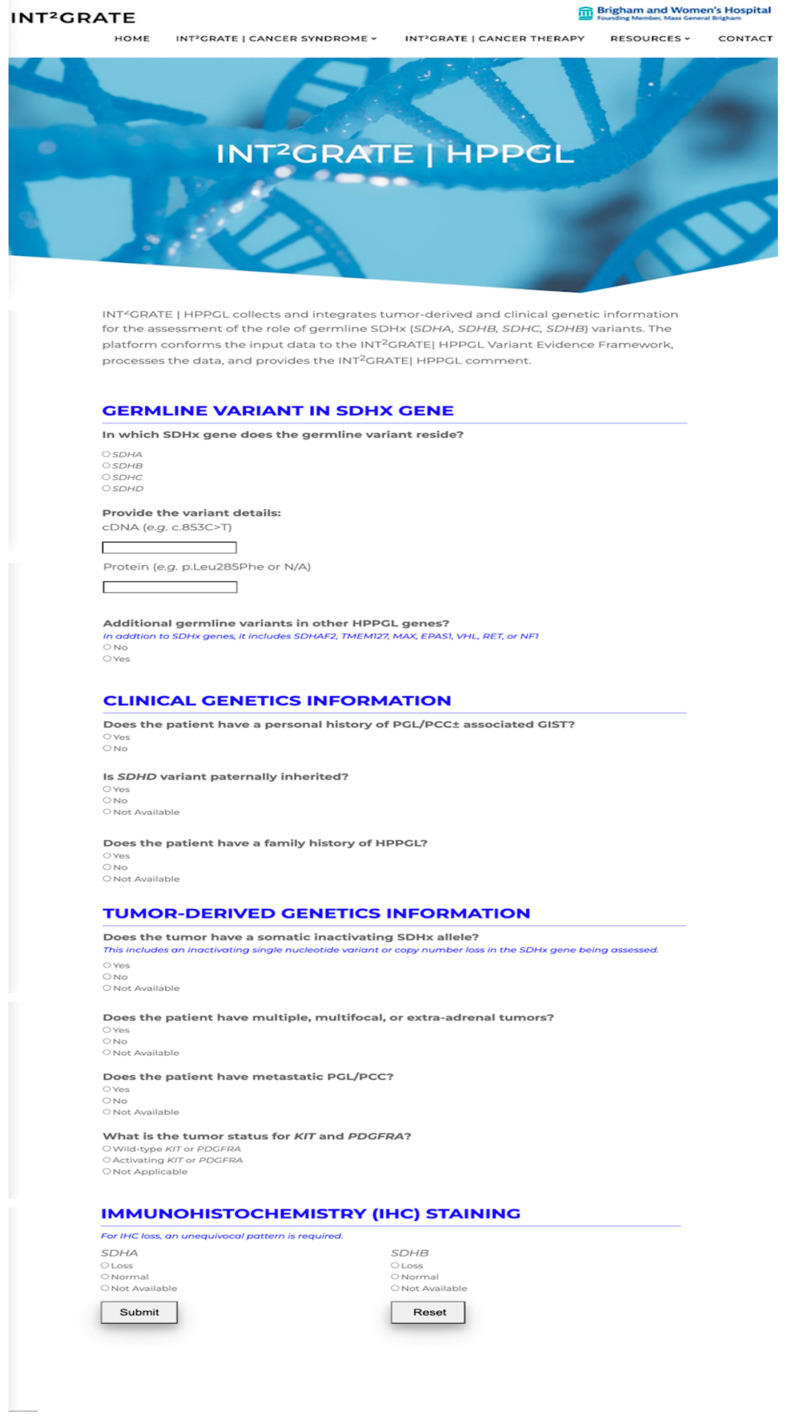
The INT^2^GRATE|HPPGL Web-Based User Interface (UI). INT^2^GRATE UI enables an intuitive collation of key evidence for the comprehensive assessment of germline variants. Upon completion of the form, the assessment is performed in the backend according to HPPGL|Variant Evidence Framework, and the INT^2^GRATE categories and associated comments are displayed.

**Table 1 cancers-16-00947-t001:** INT2GRATE|HPPGL Variant Evidence Framework.

		Germline Variant Information		Clinical Genetics Information			Tumor-Derived Information				Tumor Immunohistochemistry Staining		INT^2^GRATE Category	INT^2^GRATE Comment
	INT^2^GRATE ID Codes	SDHx Gene	Additional Germline Variants in HPPGL- Associated Genes ❖	Personal History of PGL/PCC +/− GIST Associated Tumors	Paternal Inheritance of Germline Variant	Family History of PGL/PCC /GIST	Multiple, Multifocal or Extra Adrenal Tumors	PGL/PCC Metastasis	Somatic Inactivating SDHx Allele	KIT and PDGFRA Mutation Status	SDHA	SDHB		
**SDHA TABLE**	A-I	*SDHA*	No	Yes	Yes, No, N/A	Yes, N/A	Yes, No, N/A	Yes, No, N/A	Yes	KIT and PDGFRA WT or N/A	Loss	Loss	INT^2^GRATE POSITIVE	The combination of clinical genetics evidence, tumor features, presence of somatic inactivating allele in the SDHA locus, and loss of SDHA and SDHB by immunohistochemistry support the likely relevance of this SDHA variant to Hereditary Paraganglioma-Pheochromocytoma. Correlation with clinical findings and other studies is advised.
	A-II	*SDHA*	No	Yes	Yes, No, N/A	Yes, N/A	Yes, No, N/A	Yes, No, N/A	Yes	KIT and PDGFRA WT or N/A	Normal	Loss	INT^2^GRATE NEUTRAL	Immunohistochemistry for SDH proteins shows loss of SDHB and retained SDHA expression. This pattern is supportive of a defect in one of the members of the SDH complex, but does not specifically support the presence of biallelic inactivation of SDHA. Therefore, the significance of this germline variant is uncertain. Review of the immunohistochemical findings is advised and repeat immunohistochemistry could be considered for the assessment of this variant.
	A-III	*SDHA*	No	Yes	Yes, No, N/A	Yes, N/A	Yes, No, N/A	Yes, No, N/A	Yes	KIT and PDGFRA WT or N/A	Loss	Normal	INT^2^GRATE NEUTRAL	Immunohistochemistry for SDH proteins shows an unusual staining pattern (isolated loss of SDHA). Therefore, the significance of this germline variant is uncertain. Review of the immunohistochemical findings is advised and repeat immunohistochemistry could be considered for the assessment of this variant.
	A-IV	*SDHA*	No	Yes	Yes, No, N/A	Yes, N/A	Yes, No, N/A	Yes, No, N/A	Yes	KIT and PDGFRA WT or N/A	Normal	Normal	INT^2^GRATE NEUTRAL	INT^2^GRATE requires loss of SDHA and SDHB expression by immunohistochemistry for the assessment of this variant. Without additional evidence, the significance of this germline variant in relation to Hereditary Paraganglioma-Pheochromocytoma is uncertain.
	A-V	*SDHA*	No	No	Yes, No, N/A	No	No	No	No	KIT and PDGFRA WT or N/A	Normal	Normal	INT^2^GRATE NEGATIVE	The combination of the negative personal and family history of PGL/PCC and associated tumors, absence of somatic inactivating alteration in SDHA, and retained SDHA and SDHB expression by immunohistochemistry does not support the involvement of this germline variant in Hereditary Paraganglioma-Pheochromocytoma syndromes. Correlation with clinical findings and other studies is advised.
**SDHB TABLE**	B-I	*SDHB*	No	Yes	Yes, No, N/A	Yes, N/A	Yes, N/A	Yes, N/A	Yes	KIT and PDGFRA WT	Normal	Loss	INT^2^GRATE POSITIVE	The combination of clinical genetics evidence, tumor features, presence of somatic inactivating allele in the SDHB locus, and loss of SDHB by immunohistochemistry support the likely relevance of this SDHB variant to Hereditary Paraganglioma-Pheochromocytoma. Correlation with clinical findings and other studies is advised.
	B-II	*SDHB*	No	Yes	Yes, No, N/A	Yes, N/A	Yes, No, N/A, Any	Yes, No, N/A (Any)	Yes	KIT and PDGFRA WT	Loss	Loss	INT^2^GRATE NEUTRAL	Immunohistochemistry for SDH proteins shows loss of both SDHA and SDHB, suggesting biallelic inactivation of *SDHA*. Therefore, the significance of this germline variant in relation to Hereditary Paraganglioma-Pheochromocytoma is uncertain. Review of the immunohistochemical findings is advised and repeat immunohistochemistry could be considered for the assessment of this variant.
	B-III	*SDHB*	No	Yes	Yes, No, N/A	Yes, N/A	Yes, No, N/A, Any	Yes, No, N/A (Any)	Yes	KIT and PDGFRA WT	Normal	Normal	INT^2^GRATE NEUTRAL	INT^2^GRATE requires loss of SDHB expression by immunohistochemistry for the assessment of this variant. Without additional evidence, the significance of this germline variant in relation to Hereditary Paraganglioma-Pheochromocytoma is uncertain.
	B-IV	*SDHB*	No	No	Yes, No, N/A	No	No	No	No	KIT and PDGFRA WT or NA	Normal	Normal	INT^2^GRATE NEGATIVE	The combination of the negative personal and family history of PGL/PCC and associated tumors, absence of somatic inactivating alteration in *SDHB,* and absence of SDHB deficiency status by immunohistochemistry does not support the involvement of this germline variant in Hereditary Paraganglioma-Pheochromocytoma syndromes. Correlation with clinical findings and other studies is advised.
	B-V	*SDHB*	No	No	Yes, No, N/A	No	No	No	No	KIT and PDGFRA WT or NA	N/A	N/A	INT^2^GRATE NEGATIVE	The combination of the negative personal and family history of PGL/PCC and associated tumors, and absence of somatic inactivating alteration in *SDHB* does not support the involvement of this germline variant in Hereditary Paraganglioma-Pheochromocytoma syndromes. Correlation with clinical findings, age-related penetrance and other studies is advised.
**SDHC TABLE**	C-I	*SDHC*	No	Yes	Yes, No, N/A	Yes, N/A	Yes, N/A	Yes, No, N/A (Any)	Yes	KIT and PDGFRA WT	Normal	Loss	INT^2^GRATE POSITIVE	The combination of clinical genetics evidence, tumor features, presence of somatic inactivating allele in the SDHC locus, and loss of SDHB by immunohistochemistry support the likely relevance of this SDHC variant to Hereditary Paraganglioma-Pheochromocytoma. Correlation with clinical findings and other studies is advised.
	C-II	*SDHC*	No	Yes	Yes, No, N/A	Yes, N/A	Yes, N/A	N/A	Yes	KIT and PDGFRA WT	Loss	Loss	INT^2^GRATE NEUTRAL	Immunohistochemistry for SDH proteins shows loss of both SDHA and SDHB, suggesting biallelic inactivation of *SDHA*. Therefore, the significance of this germline variant in relation to Hereditary Paraganglioma-Pheochromocytoma is uncertain. Review of the immunohistochemical findings is advised and repeat immunohistochemistry could be considered for the assessment of this variant.
	C-III	*SDHC*	No	Yes	Yes, No, N/A	Yes, N/A	Yes, N/A	N/A	Yes	KIT and PDGFRA WT	Normal	Normal	INT^2^GRATE NEUTRAL	INT^2^GRATE requires loss of SDHB expression by immunohistochemistry for the assessment of this VUS. Without additional evidence, the significance of this germline variant in relation to Hereditary Paraganglioma-Pheochromocytoma is uncertain.
	C-IV	*SDHC*	No	No	Yes, No, N/A	No	No	No	No	KIT and PDGFRA WT or NA	Normal	Normal	INT^2^GRATE NEGATIVE	The combination of the negative personal and family history of PGL/PCC and associated tumors, absence of somatic inactivating alteration in SDHC, and absence of SDHB deficiency status by immunohistochemistry does not support the involvement of this germline variant in Hereditary Paraganglioma-Pheochromocytoma syndromes. Correlation with clinical findings and other studies is advised.
**SDHD TABLE**	D-I	*SDHD*	No	Yes	Yes	Yes, N/A	Yes, N/A	N/A	Yes	KIT and PDGFRA WT	Normal	Loss	INT^2^GRATE POSITIVE	The combination of clinical genetics evidence, tumor features, parent-of-origin effect (paternal inheritance of this SDHD variant), presence of somatic inactivating allele in the SDHD locus, and loss of SDHB by immunohistochemistry support the likely relevance of this SDHD variant to Hereditary Paraganglioma-Pheochromocytoma. Correlation with clinical findings and other studies is advised.
	D-II	*SDHD*	No	Yes	Yes	Yes, N/A	Yes, N/A	N/A	Yes	KIT and PDGFRA WT	Loss	Loss	INT^2^GRATE NEUTRAL	Immunohistochemistry for SDH proteins shows loss of both SDHA and SDHB, suggesting biallelic inactivation of *SDHA*. Therefore, the significance of this germline variant in relation to Hereditary Paraganglioma-Pheochromocytoma is uncertain. Review of the immunohistochemical findings is advised and repeat immunohistochemistry could be considered for the assessment of this variant.
	D-III	*SDHD*	No	Yes	Yes	Yes, N/A	Yes, N/A	N/A	Yes	KIT and PDGFRA WT	Normal	Normal	INT^2^GRATE NEUTRAL	INT^2^GRATE requires loss of SDHB expression by immunohistochemistry for the assessment of this VUS. Without additional evidence, the significance of this germline variant in relation to Hereditary Paraganglioma-Pheochromocytoma is uncertain.
	D-IV	*SDHD*	No	No	No	No	No	No	No	KIT and PDGFRA WT or NA	Normal	Normal	INT^2^GRATE NEGATIVE	The combination of the negative personal and family history of PGL/PCC and associated tumors, absence of somatic inactivating alteration in SDHD, absence of parent-of-origin effect (paternal inheritance of this SDHD variant), and absence of SDHB deficiency status by immunohistochemistry does not support the involvement of this germline variant in Hereditary Paraganglioma-Pheochromocytoma syndromes. Correlation with clinical findings and other studies is advised.
	D-V	*SDHD*	No	No	No	No	No	No	No	KIT and PDGFRA WT or NA	NA	NA	INT^2^GRATE NEGATIVE	The combination of the negative personal and family history of PGL/PCC and associated tumors, absence of somatic inactivating alteration in SDHD, absence of parent-of-origin effect (paternal inheritance of this SDHD variant) does not support the involvement of this germline variant in Hereditary Paraganglioma-Pheochromocytoma syndromes. Correlation with clinical findings, age-related penetrance and other studies is advised.

❖ In addition in SDHx genes, germline variants are accounted for in additional HPPGL-associated genes: *SDHAF2*, *MAX*, *TMEM127*, *EPAS1*, *VHL*, *RET*, and *NF1*.

**Table 2 cancers-16-00947-t002:** Summary of demographic, personal, and familial cancer characteristics of patients in Cohort 1.

Category	Number	Percent (%)
**Total Patient Number**	109	100
Male	37	0.34
Female	72	0.66
Median age at cancer diagnosis	47 (2–84)	
**Personal Cancer History**		
HPPGL-related cancer	30	0.28
Non-HPPGL-related cancer	77	0.71
Unaffected	2	0.02
**Family** **Cancer History**		
Positive for HPPGL-related cancer	4	0.04
Negative for HPPGL-related cancer	105	0.96
**Self-Reported Ancestry**		
European	80	0.73
African/African American	5	0.05
Latino/Admixed American	3	0.03
East Asian/South Asian	6	0.06
Ashkenazi Jewish	5	0.05
Other	7	0.06
Not reported	3	0.03

**Table 3 cancers-16-00947-t003:** Summary of demographic, personal, and familial cancer characteristics of patients in Cohort 2.

Category	Number	Percent
**Total Patient Number**	494	100
Male	142	28.74
Female	352	71.26
**Personal Cancer History**		
Positive for HPPGL-related cancer	36	7.29
Negative for HPPGL-related cancer	458	92.71
**Family** **history of cancer**		
Positive for HPPGL-related cancer	12	2.43
Negative for HPPGL-related cancer	482	97.57

**Table 4 cancers-16-00947-t004:** INT2GRATE|HPPGL Variant Data for Patients with Pathogenic SDHx Variants in Cohort 1 (*n* = 37).

Subject	Germline Variant In *HPPGL* Genes		Clinical Genetics Information			Tumor-Derived Information				Tumor Immunohistochemistry Staining
	Germline SDHx Allele	Other Germline Alteration in HPPGL	Personal History of PGL/PCC ± Associated GIST	Parent-of-Origin Inheritance	Family History of PGL/PCC	Multiple, Multifocal or Extra Adrenal Tumors	PGL/PCC Metastasis	Somatic SDHx Allele	KIT and PDGFRA Mutation Status	SDHA and SDHB
1	*SDHC*:c.397C>T (p.Arg133Ter)	Neg	No (GIST)	NT	Neg	Neg	Neg	Neg	WT	Intact SDHA
2	*SDHA*:c.615T>A (p.Tyr205Ter)	Neg	No (GIST)	NT	Neg	Neg	Neg	Neg	WT	Loss of SDHB and SDHA
5	*SDHB*:c.380T>G (p.Ile127Ser)	Neg	No (GIST)	NT	Yes, PGL in mother	Neg	Neg	Neg	WT	NT
7	*SDHA*:c.1753C>T (p.Arg585Trp)	Neg	Yes	NT	Neg	Neg	Yes, PGL metastasis	SDHA:c.1534C>T (p.Arg512Ter) SDHA:c.1091T>C (p.Val364Ala)	WT	Loss of SDHB, equivocal SDHA (areas of tumor that are negative)
8	*SDHB*:c.600G>T (p.Trp200Cys)	Neg	No (GIST)	NT	Neg	Neg	Neg	Neg	WT	Loss of SDHB
11	*SDHC*:c.224G>A (p.Gly75Asp)	Neg	No (GIST)	NT	Neg	Neg	Neg	Neg	WT	NT
12	*SDHA*:c.91C>T (p.Arg31Ter)	Neg	No (GIST)	NT	Neg	Neg	Neg	Neg	WT	NT
13	*SDHC*:c.43C>T (p.Arg15Ter)	Neg	Yes	NT	Neg	Neg	Neg	Neg	WT	Intact SDHA, loss of SDHB
14	*SDHB*:c.380T>G (p.Ile127Ser)	Neg	Yes	NT	Neg	Neg	Neg	SDHB One Copy Deletion	WT	NT
15	*SDHC*:c.397C>T (p.Arg133Ter)	Neg	No (GIST)	NT	Neg	Neg	Neg	Neg	WT	Loss of SDHB
18	*SDHB*:c.605_609dupACGGA (p.Asp204fs)	Neg	No	NT	Yes, PGL in son	Neg	Neg	SDHB One Copy Deletion	WT	NT
20	*SDHC*:c.223G>C (p.Gly75Arg)	Neg	Yes	NT	Neg	Neg	Neg	Neg	WT	Intact SDHB and SDHA
22	*SDHC*:c.397C>T (p.Arg133Ter)	Neg	Yes	NT	Neg	Neg	Neg	Neg	WT	Loss of SDHB
23	*SDHC*:c.397C>T (p.Arg133Ter)	Neg	No	NT	Yes, PGL in Father	Neg	Neg	Neg	WT	Intact SDHA and SDHB
28	*SDHB*:c.72+1G>T	Neg	Yes	NT	Neg	Neg	Neg	SDHB One Copy Deletion	WT	Loss of SDHB
30	*SDHB*:c.194T>A (p.Leu65His)	Yes, SDHB:c.200+3G>C	Yes	NT	Neg	Neg	Yes, PGL metastasis	Neg	WT	Loss of SDHB
32	*SDHB*:c.137G>A (p.Arg46Gln)	Neg	No	NT	Neg	Neg	Neg	Neg	WT	NT
33	*SDHB*:c.137G>A (p.Arg46Gln)	Neg	Yes	NT	Neg	Neg	Neg	SDHB One Copy Deletion	WT	Loss of SDHB, intact SDHA
35	*SDHC*:c.397C>T (p.Arg133Ter)	Neg	No	NT	Neg	Neg	Neg	Neg	WT	SDHB is intact
37	*SDHB* EX8_3’UTR Del	Neg	Yes	NT	Neg	Neg	Neg	SDHB One Copy Deletion	WT	SDHB is intact
38	*SDHA*:c.91C>T (p.Arg31Ter)	Neg	No (GIST)	NT	Neg	Neg	Neg	SDHA One Copy Deletion	WT	NT
43	*SDHC*:c.397C>T (p.Arg133Ter)	Neg	Yes	NT	Neg	Neg	Neg	Neg	WT	SDHB is intact
46	*SDHC*:c.397C>T (p.Arg133Ter)	Neg	No	NT	Neg	Neg	Neg	Neg	WT	NT
51	*SDHA* EX1_9 Del	Neg	No (GIST)	NT	Neg	Neg	Neg	SDHA One Copy Deletion	WT	Loss of SDHB and SDHA
53	*SDHC* EX6 Del	Yes, RET:c.1771G>A	No	NT	Neg	Neg	Neg	Neg	WT	NT
54	*SDHA*:c.1794+1G>A	Yes, SDHA:c.1246A>G (p.Asn416Asp)	No	NT	Neg	Neg	Neg	Neg	WT	NT
59	*SDHC*:c.43C>T (p.Arg15Ter)	Neg	Yes	NT	Neg	Neg	Neg	Neg	WT	Loss of SDHB
60	*SDHA*:c.563G>A (p.Arg188Gln)	Neg	No	NT	Neg	Neg	Neg	SDHA Copy Number Gain (<6 copies)	WT	NT
61	*SDHA*:c.688delG (p.Glu230fs)	Neg	Yes (+GIST)	NT	Neg	Neg	Neg	SDHA:c.1432+1G>C	WT	Loss of SDHB and SDHA
63	*SDHB*:c.487T>C (p.Ser163Pro)	Yes, SDHA:c.553C>T (p.Gln185Ter) SDHA:c.622-7C>T	Yes (+GIST)	NT	Neg	Neg	Neg	SDHB One Copy Deletion SDHA:c.1043_1055del (p.Thr348LysfsTer37)	WT	Loss of SDHB
72	*SDHA*:c.91C>T (p.Arg31Ter)	Neg	No	NT	Neg	Neg	Neg	Neg	WT	NT
79	*SDHA*:c.91C>T (p.Arg31Ter)	Neg	No	NT	Neg	Neg	Neg	Neg	WT	NT
80	*SDHB* Ex1 Del	Neg	No	NT	Neg	Neg	Neg	Neg	WT	Loss of SDHB
82	*SDHB*:c.137G>A (p.Arg46Gln)	Neg	Yes (+GIST)	NT	Yes, PGL in Brother	Yes	Yes, PGL metastasis	SDHB One Copy Deletion	WT	Loss of SDHB
92	*SDHC*:c.397C>T (p.Arg133Ter)	Neg	No	NT	Neg	Neg	Neg	Neg	WT	NT
93	*SDHD*:c.106C>T (p.Gln36Ter)	Neg	Yes	SDHD variant paternally inherited	Yes, PGL in Father	Neg	Neg	SDHD One Copy Deletion	WT	Loss of SDHB
107	*SDHB*:c.418G>T (p.Val140Phe)	Neg	Yes	NT	Neg	Neg	Yes, PGL metastasis	SDHB One Copy Deletion	WT	Loss of SDHB

HPPGL: hereditary pheochromocytoma paraganglioma, PGL: paraganglioma, PCC: pheochromocytoma, GIST: gastrointestinal stromal tumor, Neg: Negative, WT: wild type, NT: not tested.

**Table 5 cancers-16-00947-t005:** INT2GRATE|HPPGL Variant Data for SDHA Analysis in Cohort 1 (*n* = 40).

Subject	Germline Variant in *HPPGL* Genes		Clinical Genetics Information			Tumor-Derived Information				Tumor Immunohistochemistry Staining
	Germline SDHx Allele	Other Germline Alteration in HPPGL	Personal History of PGL/PCC ± Associated GIST	Parent-of-Origin Inheritance	Family History of PGL/PCC/GIST	Multiple, Multifocal or Extra Adrenal Tumors	PGL/PCC Metastasis	Somatic SDHx Allele	KIT and PDGFRA Mutation Status	SDHA and SDHB
6	*SDHA*:c.853C>T (p.Leu285Phe)	Neg	No	NA	Neg	Neg	Neg	Neg	WT	NT
9	*SDHA*:c.1216G>A (p.Val406Met)	Neg	No	NA	Neg	Neg	Neg	Neg	WT	NT
17	*SDHB*:c.170A>G (p.His57Arg)	*Yes, SDHA:c.-4A,* *RET*:c.405C>T (p.Gly135=)	Yes	NA	Neg	Neg	Neg	Neg	WT	Intact SDHB
21	*SDHA:*c.1894G>T (p.Val632Phe)	Neg	No	NA	Neg	Neg	Neg	Neg	WT	NT
24	*SDHA*:c.1523C>T (p.Thr508Ile)	Neg	Yes	NA	Neg	Neg	Yes, PGL metastasis	Neg	WT	Intact SDHB
31	*SDHA*:c.596C>T (p.Ser199Leu)	Neg	No	NA	Neg	Neg	Neg	Neg	WT	NT
34	*SDHA*:c.287C>T (p.Thr96Ile)	Neg	No	NA	Neg	Neg	Neg	Neg	WT	NT
39	*SDHA*:c.133G>A (p.Ala45Thr)	Neg	No	NA	Neg	Neg	Neg	Neg	WT	NT
41	*SDHA*:c.872A>T (p.Glu291Val)	Neg	No	NA	Neg	Neg	Neg	Neg	WT	NT
44	*SDHA*:c.955A>C (p.Ile319Leu)	Neg	No	NA	Neg	Neg	Neg	Neg	WT	NT
45	*SDHA*:c.1006G>T (p.Asp336Tyr)	Neg	Yes (+GIST)	NA	Neg	Neg	Neg	*SDHA*:c.511C>T (p.Arg171Cys)	WT	Intact SDHA, intact SDHB
47	*SDHA*:c.1908+6T>C	Neg	No	NA	Neg	Neg	Neg	Neg	WT	NT
48	*SDHA*:c.464A>G (p.Asn155Ser)	Neg	No	NA	Neg	Neg	Neg	Neg	WT	NT
49	*SDHA*:c.1472A>C (p.Glu491Ala)	Neg	No	NA	Neg	Neg	Neg	Neg	KIT:c.1091A>G; PDGFRA:WT	NT
50	*SDHA*:c.530G>C (p.Ser177Thr)	Neg	No	NA	Neg	Neg	Neg	Neg	WT	NT
55	*SDHA*:c.737G>A (p.Arg246His)	Neg	No	NA	Neg	Neg	Neg	Neg	WT	NT
56	*SDHA*:c.830C>T (p.Thr277Met)	Neg	No	NA	Neg	Neg	Neg	Neg	WT	NT
57	*SDHA*:c.464A>G (p.Asn155Ser)	Neg	No	NA	Neg	Neg	Neg	Neg	WT	NT
62	*SDHA*:c.1064+5G>A	Neg	No	NA	Neg	Neg	Neg	Neg	WT	NT
65	*SDHA*:c.1064+5G>A	Neg	No	NA	Neg	Neg	Neg	Neg	WT	NT
66	*SDHA*:c.1325C>T (p.Ala442Val)	Neg	Yes	NA	Neg	Neg	Neg	Neg	WT	Loss of SDHB
67	*SDHA*:c.955A>C (p.Ile319Leu)	Neg	No	NA	Neg	Neg	Neg	Neg	WT	NT
68	*SDHA*:c.1115C>G (p.Pro372Arg)	Neg	No	NA	Neg	Neg	Neg	Neg	WT	NT
70	*SDHA*:c.1340A>G (p.His447Arg)	Neg	No	NA	Neg	Neg	Neg	Neg	WT	NT
74	*SDHA*:c.1648A>C (p.Lys550Gln)	Neg	No	NA	Neg	Neg	Neg	Neg	WT	NT
75	*SDHA*:c.1763C>T (p.Ser588Leu)	Neg	No	NA	Neg	Neg	Neg	Neg	WT	NT
81	*SDHA*:c.872A>T (p.Glu291Val)	Yes, SDHC:c.54T>G (p.Phe18Leu)	No	NA	Neg	Neg	Neg	Neg	WT	NT
84	*SDHA*:c.1835G>A (p.Gly612Glu)	Neg	No	NA	Neg	Neg	Neg	Neg	WT	NT
85	*SDHA*:c.499A>C (p.Lys167Gln)	Neg	No	NA	Neg	Neg	Neg	Neg	WT	NT
86	*SDHA*:c.613T>C (p.Tyr205His)	Neg	No	NA	Neg	Neg	Neg	Neg	WT	NT
88	*SDHA*:c.737G>A (p.Arg246His)	Neg	No	NA	Neg	*Neg*	Neg	Neg	WT	NT
89	*SDHA*:c.287C>T (p.Thr96Ile)	Neg	No	NA	Neg	Neg	Neg	Neg	WT	NT
90	*SDHA*:c.391G>A (p.Asp131Asn)	Neg	No	NA	Neg	Neg	Neg	*SDHA* Copy Number Gain(6n)	WT	NT
91	*SDHA*:c.1429C>T (p.Pro477Ser)	Neg	No	NA	Neg	Neg	Neg	Neg	WT	NT
95	*SDHA:c.1115C>G* (p.Pro372Arg)	Neg	No	NA	Neg	Neg	Neg	Neg	WT	NT
96	*SDHA*:c.830C>T (p.Thr277Met)	Yes, SDHB:c.523G>A (p.Glu175Lys)	No	NA	Neg	Neg	Neg	Neg	WT	NT
99	*SDHA*:c.935G>T (p.Arg312Leu)	Neg	No	NA	Neg	Neg	Neg	Neg	WT	NT
100	*SDHA*:c.830C>T (p.Thr277Met)	Neg	No	NA	Neg	Neg	Neg	Neg	WT	NT
101	*SDHA*:c.1908+6T>C	Neg	No	NA	Neg	Neg	Neg	Neg	WT	NT
105	*SDHA*:c.724G>A (p.Gly242Arg)	Neg	No	NA	Neg	Neg	Neg	Neg	KIT:WT; PDGFRA:c.369C>G	NT

HPPGL: hereditary pheochromocytoma paraganglioma, PGL: paraganglioma, PCC: pheochromocytoma, GIST: gastrointestinal stromal tumor, Neg: Negative, WT: wild type, NT: not tested.

**Table 6 cancers-16-00947-t006:** INT2GRATE | HPPGL Variant Data for SDHB Analysis in Cohort 1 (*n* = 15).

Subject	Germline Variant in *HPPGL* Genes		Clinical Genetics Information			Tumor-Derived Information				Tumor Immunohistochemistry Staining
	Germline SDHx Allele	Other Germline Alteration in HPPGL	Personal History of PGL/PCC ± Associated GIST	Parent-of-Origin Inheritance	Family History of PGL/PCC/GIST	Multiple, Multifocal or Extra Adrenal Tumors	PGL/PCC Metastasis	Somatic SDHx Allele	KIT and PDGFRA Mutation Status	SDHA and SDHB
3	*SDHB*:159_184delIns25 (p.Gly53fs)	Neg	Yes (+GIST)	NA	Neg	Neg	Neg	Neg	WT	NT
4	*SDHB*:c.72+3G>A	Neg	No	NA	Neg	Neg	Neg	Neg	WT	NT
10	*SDHB*:c.158G>A (p.Gly53Glu)	Neg	No	NA	Neg	Neg	Neg	Neg	WT	NT
19	*SDHB*:c.487T>C (p.Ser163Pro)	Neg	No	NA	Neg	Neg	Neg	*SDHB* One Copy Deletion	WT	Intact SDHB (IHC on RCC)
25	*SDHB*:c.478_480del (p.Lys160del)	Neg	No	NA	Neg	Neg	Neg	Neg	WT	NT
26	*SDHB*:c.553G>A (p.Glu185Lys)	Neg	No	NA	Neg	Neg	Neg	Neg	WT	NT
27	*SDHB Duplication*	Neg	No	NA	Neg	Neg	Neg	Neg	WT	NT
36	*SDHB*:c.687G>C (p.Glu229Asp)	Neg	No	NA	Neg	Neg	Neg	Neg	WT	NT
58	*SDHB*:c.79C>G (p.Arg27Gly)	Neg	No	NA	Neg	Neg	Neg	Neg	WT	NT
69	*SDHB*:c.329C>T (p.Thr110Ile)	Neg	No	NA	Neg	Neg	Neg	Neg	WT	NT
71	*SDHB*:c.317A>G (p.Asn106Ser)	Neg	No	NA	Neg	Neg	Neg	Neg	WT	NT
78	*SDHB*:c.67C>G (p.Leu23Val)	Neg	No	NA	Neg	Neg	Neg	Neg	WT	NT
83	*SDHB*:c.478_480del (p.Lys160del)	Neg	No	NA	Neg	Neg	Neg	Neg	WT	NT
97	*SDHB*:c.529C>A (p.Arg177Ser)	Neg	No	NA	Neg	Neg	Neg	Neg	WT	NT
109	*SDHB*:c.178A>G (p.Thr60Ala)	Neg	No	NA	Neg	Neg	Neg	Neg	WT	NT

HPPGL: hereditary pheochromocytoma paraganglioma, PGL: paraganglioma, PCC: pheochromocytoma, GIST: gastrointestinal stromal tumor, IHC: immunohistochemistry, Neg: Negative, WT: wild type, NT: not tested.

**Table 7 cancers-16-00947-t007:** INT2GRATE|HPPGL Variant Data for SDHC Analysis in Cohort 1 (*n* = 10).

Subject	Germline Variant in *HPPGL* Genes		Clinical Genetics Information			Tumor-Derived Information				Tumor Immunohistochemistry Staining
	Germline SDHx Allele	Other Germline Alteration in HPPGL	Personal History of PGL/PCC ± Associated GIST	Parent-of-Origin Inheritance	Family History of PGL/PCC/GIST	Multiple, Multifocal or Extra Adrenal Tumors	PGL/PCC Metastasis	Somatic SDHx Allele	KIT and PDGFRA Mutation Status	SDHA and SDHB
16	*SDHC*:c.20+6T>G	Neg	No	NA	Neg	Neg	Neg	Neg	WT	NT
52	*SDHC*:c.476C>T (p.Thr159Ile)	Neg	No	NA	Neg	Neg	Neg	Neg	WT	NT
64	*SDHC*:c.292T>G (p.Ser98Ala)	Neg	No	NA	Neg	Neg	Neg	Neg	WT	NT
77	*SDHC*:c.292T>G (p.Ser98Ala)	Neg	No	NA	Neg	Neg	Neg	Neg	WT	NT
87	*SDHC*:c.103A>G (p.Lys35Glu)	Neg	No	NA	Neg	Neg	Neg	Neg	WT	NT
94	*SDHC*:c.15G>T (p.Leu5Phe)	Neg	No	NA	Neg	Neg	Neg	Neg	WT	NT
102	*SDHC*:c.40C>T (p.Leu14Phe)	Neg	No	NA	Neg	Neg	Neg	SDHC Gain	WT	NT
103	*SDHC*:c.20+20G>T	Neg	No	NA	Neg	Neg	Neg	*SDHC Gain (estimated 6 copies)*	WT	NT
104	*SDHC*:c.32G>T (p.Arg11Leu)	Neg	No	NA	Neg	Neg	Neg	Neg	WT	NT
106	*SDHC:c.-30-?_*2318+?dup*	Neg	No	NA	Neg	Neg	Neg	*SDHC* Gain (estimated 8 copies)	WT	NT

HPPGL: hereditary pheochromocytoma paraganglioma, PGL: paraganglioma, PCC: pheochromocytoma, GIST: gastrointestinal stromal tumor, Neg: Negative, WT: wild type, NT: not tested.

**Table 8 cancers-16-00947-t008:** INT2GRATE|HPPGL Variant Data for SDHD Analysis in Cohort 1 (*n* = 7).

Subject	Germline Variant in *HPPGL* Genes		Clinical Genetics Information			Tumor-Derived Information				Tumor Immunohistochemistry Staining
	Germline SDHx Allele	Other Germline Alteration in HPPGL	Personal History of PGL/PCC ± Associated GIST	Parent of Origin Inheritance	Family History of PGL/PCC/GIST	Multiple, Multifocal or Extra Adrenal Tumors	PGL/PCC Metastasis	Somatic SDHx Allele	KIT and PDGFRA Mutation Status	SDHA and SDHB IHC
29	*SDHD*:c.453A>C (p.Lys151Asn)	Neg	No	NT	Neg	Neg	Neg	Neg	WT	NT
40	*SDHD*:c.400T>G (p.L134V)	Neg	No	NT	Neg	Neg	Neg	Neg	WT	NT
42	*SDHD*:c.53C>T (p.Ala18Val)	Neg	No	NT	Neg	Neg	Neg	Neg	WT	NT
73	*SDHD*:c.80G>A (p.Arg27Lys)	Neg	No	NT	Neg	Neg	Neg	Neg	WT	NT
76	*SDHD*:c.101T>G (p.Phe34Cys)	Neg	No	NT	Neg	Neg	Neg	Neg	WT	NT
98	*SDHD*:c.53C>T (p.A18V)	Neg	No	NT	Neg	Neg	Neg	Neg	WT	NT
108	*SDHD*:c.428A>G (p.Asn143Ser)	Neg	No	NT	Neg	Neg	Neg	Neg	WT	NT

HPPGL: hereditary pheochromocytoma paraganglioma, PGL: paraganglioma, PCC: pheochromocytoma, GIST: gastrointestinal stromal tumor, IHC: immunohistochemistry, Neg: Negative, WT: wild type, NT: not tested.

## Data Availability

The data and material of this study are available upon request from the corresponding author, A.A.G.

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
