# Peer review of "Advancing Precision Oncology in Hereditary Paraganglioma-Pheochromocytoma Syndromes: Integrated Interpretation and Data Sharing of the Germline and Tumor Genomes"

_cancers, 2024, doi:10.3390/cancers16050947_

Round 1

Reviewer 1 Report

Comments and Suggestions for Authors

The authors present the development of the novel INT²GRATE | HPPGL platform and its utility in the integrated assessment of germline and somatic mutations in pheochromocytoma and paraganglioma. Moreover, they describe the collection and processing of 8600 SDHx variants using this platform which were subsequently directly submitted to ClinVar. I agree with the authors that this novel method of processing and sharing germline and somatic mutations is crucial in advancing precision medicine and the care of oncological patients.

Author Response

Authors’ response: thank you for your review and this great summary of our work.

Reviewer 2 Report

Comments and Suggestions for Authors

The text introduces the Hereditary Paraganglioma-Pheochromocytoma Syndromes (HPPGL), a group of conditions characterized by tumors such as paragangliomas (PGLs) and pheochromocytomas (PCCs). It emphasizes the importance of understanding the genetic background of HPPGL, which affects a rare percentage of individuals. The text discusses the involvement of various genes, including SDHA, SDHB, SDHC, and SDHD, in causing HPPGL and highlights the need for a comprehensive assessment of germline variants without confounding factors.

The novel INT²GRATE platform is introduced as a multi-institutional Oncology Consortium aiming to integrate germline and tumor-derived information, ultimately advancing precision oncology in genomics.

The objective of merging data from tumours and germlines. uses of INT²GRATE for study of germline variations, immunohistochemistry patterns, and genetic data acquired from tumours. The findings of the INT²GRATE variant analysis across many patient cohorts are presented, demonstrating the platform's capacity to assess the therapeutic importance of complex variants. In-depth discussion of variant data submission to ClinVar is included in the publication, which highlights the consortium's commitment to broad genomic data sharing for improved variant interpretation and precision oncology. The paragraph underlines how crucial it is to combine information from germlines and tumours in order to increase our overall understanding of the genetic variants connected to HPPGL.

Minor revision:

Though the content seems well-written, there are a few inconsistencies that need to be fixed:

1.  Consider breaking down complex sentences into simpler ones for better clarity and comprehension.

2.  The text uses both "INT²GRATE" and "INT2GRATE" interchangeably. It would be better to stick to a consistent abbreviation throughout the text.

3. There are inconsistencies in spacing, such as "INT²GRATE | HPPGL" and "INT2GRATE | HPPGL." Consistency should be maintained.

4. The conclusion introduces the platform and consortium but does not provide a clear summary of the main findings or implications of the study. Consider summarizing the key takeaways and emphasizing the significance of the research.

Comments on the Quality of English Language

Consider breaking down complex sentences into simpler ones for better clarity and comprehension.

The text uses both "INT²GRATE" and "INT2GRATE" interchangeably. It would be better to stick to a consistent abbreviation throughout the text.

Author Response

REVIEWER #2

Comments and Suggestions for Authors

The text introduces the Hereditary Paraganglioma-Pheochromocytoma Syndromes (HPPGL), a group of conditions characterized by tumors such as paragangliomas (PGLs) and pheochromocytomas (PCCs). It emphasizes the importance of understanding the genetic background of HPPGL, which affects a rare percentage of individuals. The text discusses the involvement of various genes, including SDHA, SDHB, SDHC, and SDHD, in causing HPPGL and highlights the need for a comprehensive assessment of germline variants without confounding factors.

The novel INT²GRATE platform is introduced as a multi-institutional Oncology Consortium aiming to integrate germline and tumor-derived information, ultimately advancing precision oncology in genomics. 

The objective of merging data from tumours and germlines. uses of INT²GRATE for study of germline variations, immunohistochemistry patterns, and genetic data acquired from tumours. The findings of the INT²GRATE variant analysis across many patient cohorts are presented, demonstrating the platform's capacity to assess the therapeutic importance of complex variants. In-depth discussion of variant data submission to ClinVar is included in the publication, which highlights the consortium's commitment to broad genomic data sharing for improved variant interpretation and precision oncology. The paragraph underlines how crucial it is to combine information from germlines and tumours in order to increase our overall understanding of the genetic variants connected to HPPGL.

 Authors’ response: thank you for your review and this great summary of our work.

Minor revision:

Though the content seems well-written, there are a few inconsistencies that need to be fixed:

  1.  Consider breaking down complex sentences into simpler ones for better clarity and comprehension.

Authors’ response: We have broken down long or complex sentences to address the reviewer’s comment.

  1.  The text uses both "INT²GRATE" and "INT2GRATE" interchangeably. It would be better to stick to a consistent abbreviation throughout the text.

Authors’ response: thank you for catching that. INT²GRATE is now correctly displayed throughout the manuscript and supplementary materials.  

  1. There are inconsistencies in spacing, such as "INT²GRATE | HPPGL" and "INT2GRATE | HPPGL." Consistency should be maintained.

Authors’ response: it looks like after fixing “INT²GRATE”, spacing inconsistencies in INT²GRATE | HPPGL is resolved. Thank you for pointing this out.

  1. The conclusion introduces the platform and consortium but does not provide a clear summary of the main findings or implications of the study. Consider summarizing the key takeaways and emphasizing the significance of the research.

Authors' response: A summary of the implications of the study is added as suggested by the reviewer.

Comments on the Quality of English Language

Consider breaking down complex sentences into simpler ones for better clarity and comprehension.

Authors’ response: addressed under “Minor revision 1” above.

The text uses both "INT²GRATE" and "INT2GRATE" interchangeably. It would be better to stick to a consistent abbreviation throughout the text.

Authors’ response: addressed under “Minor revision 2” above.

Reviewer 3 Report

Comments and Suggestions for Authors

The article "Advancing Precision Oncology in Hereditary Paraganglioma-2 Pheochromocytoma Syndromes: Integrated Interpretation and Data Sharing of the Germline and Tumor Genomes" discusses the use of the INT²GRATE platform for the analysis of genetic variants in hereditary paraganglioma-2 pheochromocytoma syndromes. The platform integrates clinical genetics, germline variants, and tumor-derived genetic data to create a comprehensive database. It also facilitates data sharing and variant submission to ClinVar. However, the study focuses on specific hereditary syndromes and may not cover all genetic mechanisms and clinical presentations associated with pheochromocytoma and paraganglioma.

The article presents a comprehensive approach to understanding and managing hereditary paraganglioma-2 pheochromocytoma syndromes. However, like any scientific research, it has its limitations.

1. Dependence on User Input: The INT²GRATE platform, a significant part of the study, relies heavily on user input. The accuracy of the results is contingent on the accuracy of the information the user provides. Suppose the user's combination of responses is outside scenarios accounted for in the INT²GRATE Variant Evidence Framework. In that case, a message will notify the user that their combination of responses is not currently accounted for within the Variant Evidence Framework.

2. Limited Scope of Genetic Variants: The study focuses on specific germline variants in the SDHx gene and other HPPGL genes. However, the genetic landscape of hereditary paraganglioma-2 pheochromocytoma syndromes is complex and continually evolving. New genetic variants and mechanisms of tumorigenesis are being discovered, which may need to be covered by the current framework.

3. Reliance on Immunohistochemistry: The study uses immunohistochemistry for SDHB and SDHA proteins as part of its evidence framework. While this is a standard procedure, it may only capture some of the molecular complexities of the disease.

4. Data Sharing and Privacy Concerns: The platform facilitates data sharing via the ClinVar API, which could raise concerns about patient privacy and data security.

5. Lack of Clinical Validation: While the study presents a comprehensive framework for understanding and managing hereditary paraganglioma-2 pheochromocytoma syndromes, the approach needs to be validated more clinically. The framework's effectiveness in improving patient outcomes must be validated in clinical trials.

6. Limited Applicability to Rare Variants: The study's approach may not apply to rare or novel disease variants requiring different diagnostic and therapeutic strategies.

7. Technical Complexity: The INT²GRATE platform, while powerful, may require a certain degree of technical knowledge to use effectively. This could limit its accessibility to users without a strong technical background.

8. Inherent Limitations of Immunohistochemistry: Immunohistochemistry, while a valuable tool, has its limitations. It may not capture all the molecular complexities of the disease, and the interpretation of results can be influenced by factors such as protein life-cycle, use of evidence-based methods, and data normalization.

9. Potential for Misclassification of Variants: The study's approach may lead to the misclassification of specific genetic variants. For example, variants of uncertain significance (VUS) in the SDHx genes may be misclassified as likely benign or likely pathogenic based on the evidence framework, which could impact the interpretation of results and subsequent clinical management.

10. Lack of Consideration for Other Hereditary Syndromes: The study focuses on hereditary paraganglioma-2 pheochromocytoma syndromes, but there are other hereditary syndromes associated with pheochromocytoma and paraganglioma, such as Multiple Endocrine Neoplasia type 2 (MEN2). These syndromes may have different genetic mechanisms and clinical presentations, which are not covered by the current study.

11. Platform Independence Challenges: While the INT²GRATE platform is designed to be platform-independent, it can introduce its challenges. These include the need for more extensive testing across different platforms, potential performance issues due to the lack of a larger and more complex runtime, and the potential to be less feature-rich due to the need to accommodate a wide range of platforms.

In conclusion, while the study provides a valuable framework for understanding and managing hereditary paraganglioma-2 pheochromocytoma syndromes, it has limitations that must be considered when interpreting and applying its findings in clinical practice.

Comments on the Quality of English Language

minor

Author Response

REVIEWER #3

Comments and Suggestions for Authors

The article "Advancing Precision Oncology in Hereditary Paraganglioma-2 Pheochromocytoma Syndromes: Integrated Interpretation and Data Sharing of the Germline and Tumor Genomes" discusses the use of the INT²GRATE platform for the analysis of genetic variants in hereditary paraganglioma-2 pheochromocytoma syndromes. The platform integrates clinical genetics, germline variants, and tumor-derived genetic data to create a comprehensive database. It also facilitates data sharing and variant submission to ClinVar. However, the study focuses on specific hereditary syndromes and may not cover all genetic mechanisms and clinical presentations associated with pheochromocytoma and paraganglioma.

The article presents a comprehensive approach to understanding and managing hereditary paraganglioma-2 pheochromocytoma syndromes. However, like any scientific research, it has its limitations.

  1. Dependence on User Input: The INT²GRATE platform, a significant part of the study, relies heavily on user input. The accuracy of the results is contingent on the accuracy of the information the user provides. Suppose the user's combination of responses is outside scenarios accounted for in the INT²GRATE Variant Evidence Framework. In that case, a message will notify the user that their combination of responses is not currently accounted for within the Variant Evidence Framework.

Authors’ response: we agree that the accuracy of data is indeed crucial.

Regarding the accuracy of data submission: 1) large-scale INT²GRATE variant submission via the API is only performed by the INT²GRATE coordinating center, with built-in QC steps to account for the accuracy of these large data submissions. Also, 2) submission of one variant at a time by the UI is only open to the registered users (current and future members of the consortium) after review by the INT²GRATE expert panel.

Regarding the accuracy of result generation: The combination of scenarios outlined in the INT²GRATE Variant Evidence Framework is hard coded in the backend. The backend logic is also coded to identify scenarios that are outside the Variant Evidence Framework to account for complexities explained in the manuscript. In these cases, a message is returned to indicate the scenario is not part of the INT²GRATE | HPPGL Variant Evidence Framework. This ensures the user will receive accurate information and that uninformative scenarios are not submitted to ClinVar.

  1. Limited Scope of Genetic Variants: The study focuses on specific germline variants in the SDHx gene and other HPPGL genes. However, the genetic landscape of hereditary paraganglioma-2 pheochromocytoma syndromes is complex and continually evolving. New genetic variants and mechanisms of tumorigenesis are being discovered, which may need to be covered by the current framework.

Authors' response: INT²GRATE | HPPGL is only intended for HPPGL caused by mutations in the SDHx for the detailed reasons explained in the rationale in the manuscript. New or future HPPGL genetic mechanisms can be covered in future versions of the platform.

To address the reviewer’s comment, we added in the limitation section in the discussion that future studies will address or incorporate new HPPGL mechanisms in the platform.

  1. Reliance on Immunohistochemistry: The study uses immunohistochemistry for SDHB and SDHA proteins as part of its evidence framework. While this is a standard procedure, it may only capture some of the molecular complexities of the disease.

Authors' response: We agree that the reliance on IHC has limitations and may not capture some complexities of HPPGL. To minimize the impact of this limitation, we use a comprehensive set of evidence, not just IHC, in the evidence framework. To address reviewers' comments, we expanded on the IHC limitations in the limitation section of the discussion.

  1. Data Sharing and Privacy Concerns: The platform facilitates data sharing via the ClinVar API, which could raise concerns about patient privacy and data security.

Authors' response: Variant submission via the ClinVar API is only done by the INT2GRATE consortium submitter process. This process is reviewed and approved by our institution's security and privacy guidelines. No protected health information (PHI) is shared or submitted in this process. To clarify this point, we have added this information to section 4.2.

  1. Lack of Clinical Validation: While the study presents a comprehensive framework for understanding and managing hereditary paraganglioma-2 pheochromocytoma syndromes, the approach needs to be validated more clinically. The framework's effectiveness in improving patient outcomes must be validated in clinical trials.

Authors’ response: We agree with the reviewer, we provide a framework for better understanding specific variants that may confer risk for HPPGL syndromes. However, management of HPPGL syndromes is beyond the scope of this work. Notably, the management of HPPGL is widely different, and currently, consensus and guidelines are being developed (as published in the International Consensus Guidelines). and validated. To clarify this further, additional sentences and a reference to this point are added to the first paragraph on page 11, and to the concluding remarks.

  1. Limited Applicability to Rare Variants: The study's approach may not apply to rare or novel disease variants requiring different diagnostic and therapeutic strategies.

Authors' response: we believe that this study is specifically well suited for rare SDHx variants. Many rare or novel SDHx variants are classified as VUS due to a lack of available clinical and molecular data for the variants. INT²GRATE | HPPGL collects comprehensive clinical and molecular data about these rare variants and promotes the sharing of this data, which is particularly important for rare variants. Variants in other PGL-PCC genes (outside of the SDHx genes) and different diagnostic and therapeutic strategies are outside the scope of this study but can be addressed in future versions of the platform.

To address the reviewer’s comment we have added the limitation on novel disease strategies in the limitation paragraph in the discussion.

  1. Technical Complexity: The INT²GRATE platform, while powerful, may require a certain degree of technical knowledge to use effectively. This could limit its accessibility to users without a strong technical background.

Authors’ response: The INT²GRATE UI was developed to address this very point. The user requires no technical knowledge. It allows the registered user with only internet access to access the webpage and simply answer the radio button questionnaire and have the relevant comments to show. All backend coding and logic for the UI are developed and maintained by our institution’s INT²GRATE team. The submission of variants through ClinVAr API, which requires computational expertise, is handled by our institution’s INT²GRATE team.

  1. Inherent Limitations of Immunohistochemistry: Immunohistochemistry, while a valuable tool, has its limitations. It may not capture all the molecular complexities of the disease, and the interpretation of results can be influenced by factors such as protein life-cycle, use of evidence-based methods, and data normalization.

Authors’ response: we agree that IHC has inherent limitations. To address the reviewer’s comment, additional sentences on the limitation of IHC related to HPPGL molecular complexities have been added to the second paragraph on page 11.

  1. Potential for Misclassification of Variants: The study’s approach may lead to the misclassification of specific genetic variants. For example, variants of uncertain significance (VUS) in the SDHx genes may be misclassified as likely benign or likely pathogenic based on the evidence framework, which could impact the interpretation of results and subsequent clinical management.

Authors’ response: INT2GRATE variant collection overtime may help with variant reclassifications. Having said that we have made the point that the goal is not variant reclassification, but instead to provide additional evidence and insight for variants (paragraph 3 on page 1): “INT2GRATE | HPPGL is not intended to re-classify variants or replace the ACMG assessment criteria.  Instead, it is developed to serve as a companion tool to help genetic professionals collect and assess a comprehensive germline and associated tumor set of evidence. Through large-scale variant analysis and data sharing, the true clinical significance of complex variants can be elucidated.”

To address the reviewer’s comment, we have added sentences in the conclusion paragraph of the manuscript to clarify this further.

  1. Lack of Consideration for Other Hereditary Syndromes: The study focuses on hereditary paraganglioma-2 pheochromocytoma syndromes, but there are other hereditary syndromes associated with pheochromocytoma and paraganglioma, such as Multiple Endocrine Neoplasia type 2 (MEN2). These syndromes may have different genetic mechanisms and clinical presentations, which are not covered by the current study.

Authors' response: We agree that other genes may cause overlapping features with HPPGL. INT2GRATE evidence framework is only developed for HPPGL caused by SDHx variants, as explained in the rationale sections (explained why MEN2 and other genes are excluded). All other genes that can cause overlapping features are intentionally excluded from this INT2GRATE evidence framework and may be covered in future studies.

  1. Platform Independence Challenges: While the INT²GRATE platform is designed to be platform-independent, it can introduce its challenges. These include the need for more extensive testing across different platforms, potential performance issues due to the lack of a larger and more complex runtime, and the potential to be less feature-rich due to the need to accommodate a wide range of platforms.

  Authors' response:

  • The runtime of the algorithm is negligible, and results are returned nearly instantaneously to the user. This is because the Javascript algorithm that supports the Variant Evidence Framework is processed on the client side (i.e. meaning the user’s browser). So after the page is loaded, it should be able to support almost instantaneous processing of the result without any run time considerations. Each scenario has been hardcoded so that specific combinations of responses can return only one result corresponding to an INT²GRATE Variant Evidence Framework. For any scenario outside the current Variant Evidence Framework, the platform is coded to issue a response that indicates the combination falls outside of the current framework. Therefore, no heavy or complex computational methods are anticipated to run in real time.

  • INT²GRATE runs Variant Evidence Frameworks for different syndromes (this manuscript only addresses HPPGL). Each of the algorithms is built to be independent of the other. If the algorithm for one syndrome is updated, it will not impact the function of the algorithms for another syndrome. This feature is designed intentionally to allow each page/algorithm to be updated in isolation during any revisions. During any such revisions, a message on the page will show that the page is under maintenance.

  • The INT2GRATE coordinating center computation team performs all QC tests as part of the routine process. We do not anticipate the QC testing to create challenges for the user.

In conclusion, while the study provides a valuable framework for understanding and managing hereditary paraganglioma-2 pheochromocytoma syndromes, it has limitations that must be considered when interpreting and applying its findings in clinical practice.

Authors' response: thank you for taking the time to address these limitations. We address each of them as noted above.

Reviewer 4 Report

Comments and Suggestions for Authors

here, the Authors present the development and application of the INT²GRATE platform, using HPPLG as a case study.

While the approach is interesting, I feel that it can be much better presented in a more specialized journal. Should the Authors pursue publication in Cancers, then the clinical relevance of the findings should be extensively clarified.

Comments on the Quality of English Language

No special concerns

Author Response

REVIEWER #4

Comments and Suggestions for Authors

here, the Authors present the development and application of the INT²GRATE platform, using HPPLG as a case study.

While the approach is interesting, I feel that it can be much better presented in a more specialized journal. Should the Authors pursue publication in Cancers, then the clinical relevance of the findings should be extensively clarified.

Authors' response: we agree that the manuscript is a specialized topic intended for readers with proficient expertise in both clinical genetics and tumor aspects of the disease. This manuscript is an invited submission to a Special Topic that was created by the senior author of this manuscript on the “Clinical Applications of Integrative Analysis of Validated Somatic and Germline Genomic Results in Oncology”. Therefore, we believe that the manuscript will be appropriate for the target audience.

Round 2

Reviewer 4 Report

Comments and Suggestions for Authors

thanks for having considered my comment. I understand that you wish to pursue publication in a special issue of Cancers, of which the senior Author is the leading Editor.

In a constructive intent, and to limit the bias of submitting a paper to a special issue led by the Authors themselves, I encourage you to revise the paper and better clarify the clinical relevance of the findings of your analysis. 

Comments on the Quality of English Language

Minor polishing required.

Author Response

We appreciate your comment. We are hoping that submitting the manuscript to this focused topic (integration of germline and tumor) helps focused cancer readers learn about our proof-of-concept method for these rare tumor types. By doing so, we are also aiming to solicit more articles around the integration of somatic and germline data. Given that more labs are nowadays doing tumor and germline sequencing, this aim seems more achievable than ever.

To address your comment, and to clarify the clinical utility of our proof-of-concept manuscript further, we have added statements on the clinical relevance of our work both to the end of the introduction and the conclusion of the manuscript.